# The Impact of Whole-Body Vibration Training on Bone Minerals and Lean Mass in Children and Adolescents with Motor Disabilities: A Systematic Review and Meta-Analysis

**DOI:** 10.3390/children9020266

**Published:** 2022-02-15

**Authors:** Shuoqi Li, Wenbing Yu, Wei Li, Juncheng Wang, Lili Gao, Shiming Li

**Affiliations:** 1School of Health Science, Universiti Sains Malaysia, Kota Bharu 15000, Malaysia; lsq728738864@gmail.com; 2Institute of Sports Science, Ocean University of China, Qingdao 266100, China; yuwenbing@ouc.edu.cn (W.Y.); m18369178928@outlook.com (J.W.); haiyanglishiming@163.com (S.L.); 3College of Physical Education and Health Sciences, Zhejiang Normal University, Jinhua 321001, China; ty1986@zjnu.edu.cn; 4Department of Neurology, Qingdao Hospital of Traditional Chinese Medicine (Qingdao Hiser Hospital), Qingdao 266034, China

**Keywords:** whole-body vibration training, bone mineral, motor disabilities, children

## Abstract

Whole-body vibration training (WBVT) offers a potential auxiliary treatment method for the rehabilitation of motor disabilities to address a reduction in bone minerals and lean mass caused by motor-disability rehabilitation. The aim of this review was to analyze the efficacy of WBVT in muscle–bone rehabilitation. In order to investigate the potential effect of WBVT on children and adolescents with motor disabilities, a meta-analysis was carried out. From January 2006 to June 2021, studies that met certain criteria were searched for in the Scopus, PubMed, Web of Science, and EBSCO databases. An analysis of standardized mean differences was performed using the STATA 15.1 software with a 95% confidence interval (PROSPERO registration number: CRD42021258538). Eight studies were selected that included 179 male and 139 female children and adolescents suffering from motor disabilities. The results of the meta-analysis showed that WBVT significantly improved femur bone-mineral density ((*p* < 0.01, z = 2.66), standardized mean difference (SMD) (95% CI) = 0.41 (0.11, 0.72)), total body–bone mineral content ((*p* < 0.01, z = 3.08), SMD (95% CI) = 0.26 (0.10, 0.43)), and lean mass ((*p* < 0.01, z = 2.63), SMD (95% CI) = 0.22 (0.06, 0.39)). In addition, there was no significant effect of WBVT on lumbar spine bone mineral density in the disabled children and adolescents ((*p* = 0.21, z = 1.25), SMD (95% CI) = 0.17 (−0.10, 0.43)). WBVT can improve femur bone density, total body bone mineral content, and lean mass in children and adolescents suffering from motor disabilities, while there is no effect on lumbar-spine bone density. WBVT can be used as a potential program to improve bone minerals in children and adolescents with motor disabilities.

## 1. Introduction

Cerebral palsy and osteogenesis imperfecta are among the diseases that can cause varying degrees of motor disabilities in children and adolescents. Common features of these diseases include muscle dysfunction as well as decreased muscle and bone mass [1,2]. Muscles and bones are important body components that serve distinct and complementary roles. A reduced density of bone minerals increases the possibility of fractures, while an increased muscle mass and activity can substantially reduce this risk [2,3]. However, because traditional exercise is difficult for children with motor disabilities, developing a safe and effective rehabilitation program has become a major challenge.

For children and adolescents suffering from compromised biomechanical functions, whole-body vibration training (WBVT) has received increasing interest as a promising therapeutic method for improving the functions of lower-limb muscles and bone mass [4,5,6]. WBVT comprises standing on a vibrating plate either hands-free, holding on to a support, or in conjunction with additional devices [4]. The intensity of the vibrations can be further modified using the following parameters: frequency, amplitude, acceleration, and type of vibration plate [7]. WBVT has been postulated to enhance mechanical stress and activate proprioceptive spinal circuits, causing reactions that trigger muscle spindles and Golgi tendon organs, resulting in improved muscular function [8]. The increased mechanical load has also been considered to elicit bone adaptation through various pathways, many of which are assumed to be independent of a muscle response [9].

In a previous meta-analysis [6], the impact of vibration training on the bone mineral densities of children that have cerebral palsy was initially explored. However, only two studies were included, and the testing methods related to bone mineral indicators were not uniform. In addition, Chen et al. [10] performed a meta-analysis comparing the impact of WBVT on the lean masses of children who were thermally injured or had Down syndrome. It similarly included only two studies with a high heterogeneity and low stability of results. So far, the impact of WBVT on the bone minerals and lean body masses of children and adolescents with mobility impairments remains unknown. Although studies have shown that WBVT significantly increases the bone mineral contents and lean body masses of children with motor disabilities [11,12], there is also one study that demonstrates opposing results [13]. This study evaluated the impact of WBVT on bone minerals and lean mass through a meta-analysis, as well as provided references for the recovery of children and adolescents with motor disabilities.

## 2. Materials and Methods

### 2.1. Protocol and Registration

The review protocol was registered with the International Prospective Register of Systematic Reviews on 30 June 2021 (registration number: CRD42021258538).

### 2.2. Data Sources and Study Selection

The Scopus, PubMed, Web of Science, and EBSCO databases were searched from January 2006 to June 2021, with the final day of retrieval being 3 June 2021. The terms “whole-body vibration,” “WBV,” “training,” “exercise,” “motor disabilities”, “children,” and “adolescents” were applied during the database searches. The search strategy and results for every database are described in Appendix A. Two investigators independently evaluated the selected titles and abstracts. Following that, entire papers were assessed using inclusion and exclusion criteria (2.3). If any disputes arose among the two investigators, a third investigator was required to reach a consensus. Figure 1 displays the article selection process. This systematic review and meta-analysis followed the recommendations proposed by the Preferred Reporting Items for Systematic Reviews and Meta-Analyses (PRISMA).

### 2.3. Inclusion and Exclusion Criteria

(1) Participants: Ages 3 to 18; (2) intervention: whole-body vibration training; (3) conditions: motor disabilities; (4) outcomes: bone minerals and lean body mass; (5) study types: randomized controlled trials and cohort studies; (6) other: the outcome index was tested using dual-energy X-ray absorptiometry. Full-text papers in English. Gray literature, including abstracts, conference proceedings, and poster presentations, was not considered.

### 2.4. Quality Assessment

The quality of the selected randomized controlled trial (RCT) studies was assessed using the PEDro scale [14], which consisted of 11 questions. A score of 6 or higher indicated a high quality, while a score below 6 indicated a low quality [15]. In order to examine the quality of the included non-RCT studies, the Quality Assessment Tool for Before–After (Pre–Post) Studies with No Control Group (NIH) scale [16] (available online: https://www.nhlbi.nih.gov/health-topics/study-quality-assessment-tools (accessed on 9 March 2019)) was used, with the NIH consisting of 12 questions. The quality of every study was evaluated as “poor,” “fair,” or “good.” Levels of evidence were assessed using the Levels of Evidence for Therapeutic Studies tool (from the Centre for Evidence-Based Medicine, available online: http://www.cebm.net, Accessed on 9 March 2021). This tool was divided into 10 levels [17]. Levels of recommendations were assessed using the Grade Practice Recommendations tool (from the American Society of Plastic Surgeons. Evidence-based clinical practice guidelines, available online: http://www.plasticsurgery.org/Medical_Professionals/Health_Policy_and_Advocacy/Health_Policy_Resources/Evidence-based_GuidelinesPractice_Parameters/Description_and_Development_of_Evidence-based_Practice_Guidelines/ASPS_Grade_Recommendation_Scale.html (accessed on 3 March 2011) [17].

### 2.5. Risk of Bias Assessment

The sensitivity analysis was carried out by excluding each study at a time to determine the stability of the findings of the meta-analysis. Funnel plots were used for the analysis of the study’s publication bias.

### 2.6. Data Analysis

In order to perform the meta-analysis, STATA 15.1 (version 21.0, College Station, TX, USA) was used to input every applicable outcome variable. Although the included studies had continuous outcome variables, the test units and equipment differed. As a result, standardized mean difference (SMD) was chosen as the index of effect scale. The I^2^ statistic was employed to assess study heterogeneity, with no heterogeneity between the studies determined if I^2^ was less than 50%. Therefore, the analysis could be performed with a fixed-effect model. However, an I^2^ value that was equal to or more than 50% indicated heterogeneity between the studies. Therefore, the analysis needed to be performed with a random-effect model.

## 3. Results

### 3.1. Eligibility of Studies

Eight studies evaluated the impact of WBVT on bone minerals and lean mass in children and adolescents suffering from motor impairments; three were RCT studies and five were non-RCT studies (Table 1). All the included studies had only a baseline and final data after the intervention. The studies were ethically approved by their respective institutions, and the two researchers had a Cohen’s kappa coefficient of 0.885. The eight studies included 179 males and 139 females, with four studies reporting femoral bone mineral density and six studies reporting the bone mineral density in the lumbar spine, total-body bone content, and lean mass. There was a minimum of 12 weeks and a maximum of 6 months for the intervention period. The vibration frequency was 5–42 Hz.

### 3.2. Quality Assessment

The total scores of the PEDro scale were all higher than five points in the included RCT studies [12,13,18] and were defined as high quality. On the NIH scale, the overall quality rating of all the included non-RCT studies [11,19,20,21,22] was “good” (Table 2). For the level of evidence, three studies were a 1B level and five studies were a 2B level. For the level of recommendation, five studies were a B level and three studies were a C level.

### 3.3. Quantitative Synthesis

The effect of WBVT on femur and lumbar spine bone mineral density in children and adolescents suffering from motor disabilities was evaluated in four and six studies, respectively (*n* = 85 and 112). The meta-analysis results revealed that WBVT significantly improves femur bone-mineral density in children and adolescents suffering from motor disabilities (*p <* 0.01, z = 2.66) and SMD (95% CI) = 0.41 (0.11, 0.72) (Figure 2a). In addition, no significant heterogeneity between the studies was detected (*p* = 0.42, I^2^ = 0.0%). There was no significant effect of WBVT on lumbar spine bone mineral density in children and adolescents with motor disabilities (*p* = 0.21, z = 1.25) and SMD (95% CI) = 0.17 (−0.10, 0.43) (Figure 2b). Furthermore, no heterogeneity was observed across the studies (*p* = 0.79, I^2^ = 0%).

The impact of WBVT on the total body bone mineral content and lean mass in children and adolescents with motor disabilities (*n* = 285) was evaluated in six studies. The meta-analysis results revealed that WBVT significantly improves total body bone mineral content in children and adolescents with motor disabilities (*p <* 0.01, z = 3.08), SMD (95% CI) = 0.26 (0.10, 0.43) (Figure 3a). In addition, no significant heterogeneity between the studies was detected (*p* = 0.89, I^2^ = 0.0%). Similarly, WBVT significantly improves lean mass in children and adolescents suffering from motor impairments (*p <* 0.01, z = 2.63), SMD (95% CI) = 0.22 (0.06, 0.39) (Figure 3b). Furthermore, no heterogeneity was observed across the studies (*p* = 0.96, I^2^ = 0%).

### 3.4. Sensitivity Analysis

The sensitivity analysis was performed by modifying the analysis model, selecting the effect size, and excluding particular studies. Even though no heterogeneity was detected across all the studies, the sensitivity analysis was still carried out to ensure data accuracy and stability. The results revealed that each of the included studies had a high degree of agreement with the centerline. The combined effect size did not substantially change the association between bone mineral and lean mass despite the deletion of each study, demonstrating the excellent stability of this study. The sensitivity analyses for femur bone mineral density, lumbar spine bone mineral density, total body bone mineral content, and lean mass are displayed in Figure 4a,b and Figure 5a,b, respectively.

### 3.5. Analysis of Publication Bias

Only eight studies were selected for analysis as the sample size was relatively small for using WBVT on children and adolescents with motor disabilities. Despite using an overall sample size that was almost the minimum required for a funnel plot analysis, the analysis was still considered suitable for reflecting a publication bias to a certain extent, with minor risks. The effectiveness of using a funnel analysis for investigations with small sample sizes has been previously demonstrated [23]. Figure 6 and Figure 7 illustrate the funnel charts of WBVT’s impact on children and adolescents with motor disabilities.

## 4. Discussion

This meta-analysis aimed to examine the impact of WBVT on bone minerals and lean mass in children and adolescents suffering from motor impairments. There were eight articles included, four of which included femur bone mineral densities as outcome indicators, while six included lumbar spine bone mineral density, total body bone mineral content, and lean mass as outcome indicators. The results revealed that WBVT significantly increases femur bone density, total body bone mineral content, and lean mass in children and adolescents suffering from motor impairments. However, WBVT does not significantly affect lumbar spine bone density. These findings demonstrate that WBVT offers an efficient rehabilitation program for improving the conditions of children and adolescents with motor disabilities. Among the eight articles included, the evidence level and recommendation level are medium, and there is still a lack of high-level research.

As a new rehabilitation program, WBVT has been used to minimize defects and movement restrictions in children and adolescents [24]. It is a type of training that utilizes high-frequency mechanical stimuli produced by a vibrating platform, which is transmitted throughout the body to the load bone and to activate sensory receptors [25]. It was previously shown that WBVT improves muscle strength, lean mass, bone mass, and gross motor performance [26,27,28]. Although physical therapists worldwide have used WBVT in clinical practice, its impact on children and adolescents suffering from motor impairments remains unknown due to previous studies’ lack of methodological quality [29,30,31,32]. An investigation was previously conducted on the impact of WBVT on the health-related physical fitness of children and adolescents with disabilities [4]. In the study, it was revealed that WBVT could be beneficial for the health of disabled children and adolescents; however, low-quality studies (i.e., not randomized controlled trials) were included, the risk of a bias in the included trials was not analyzed, and a meta-analysis was not performed [4]. The dual-energy X-ray was found to be an effective index for evaluating body composition, with a high accuracy and repeatability. This study included all bone minerals and lean-mass indicators evaluated by dual-energy X-rays in order to improve the homogeneity of the meta-analysis and assess the impact of WBVT on body composition more efficiently.

Lean mass was found to be directly connected to muscle development and physical activity [33]. According to the results of this study, there is a significantly increased lean mass among children and adolescents suffering from motor impairments after WBVT. A study by Kruse et al. demonstrated a significantly increased musculus vastus lateralis thickness in children with cerebral palsy (CP) following an eight week home based progressive resistance training program, which was consistent with our findings [34]. Several studies also reported that vibration-assisted training increases the muscle masses of children with CP [18,28]. The stimulation of muscle spindles by vibration, leading to repetitive muscle contractions and training, is considered to be the physiological basis for vibration-assisted training’s muscle anabolic effect [35,36]. However, there are also studies with different points of view. In a meta-analysis by Saquetto [5], there were no significant improvements in muscle strength after WBVT compared to a control group, which may have been due to different indicators or test methods. In addition, a comparison of the muscle strengths showed that the *p* value was equal to 0.06, which was closer to the critical value of significance, potentially leading to false negatives due to a small sample size (*n* = 3).

Bone fragility in patients suffering from motor disabilities was linked to an increased rate of fragility fractures, particularly in children and adolescents [37]. The results showed that WBVT significantly improves femur bone mineral density and total body bone mineral content in children and adolescents suffering from motor impairments, but not lumbar spine bone mineral density. This may have been due to the different vibration intensities received by the different body parts. There was evidence in a study that there is an enhanced transmission of floor-based vibration at the distal tibia but a reduction at the distal femur [38]. Similarly, a previous report demonstrated significantly increased lower-limb lean mass and total body bone mineral content after a 20-week WBVT program using an identical WBVT plate from the same manufacturer of the plate used in the former study (*n* = 40) [18]. The training routine included three 3 min training sessions four times per week for 20 weeks. Overall, the participants received vibration-assisted training for 720 min at a frequency of 20 Hz and an amplitude of 1 mm. However, the Z-scores were not evaluated and a control group was not included. Another randomized controlled study investigated the impact of WBVT on bone mineral density in children with CP (*n* = 20) [13] using an identical WBVT system from the same manufacturer. A school physiotherapy program was provided to both the treatment and control groups. For the treatment group, an additional three 3 min sessions of vibration-assisted physiotherapy five times per week was provided. The treatment group received vibration-assisted training for 737–914 min at a frequency of 18 Hz and an amplitude of 4 mm (an acceleration of ~28 m/s^2^). The additional WBVT did not have any effect on bone mineral (distal femur and lumbar spine) densities in the treatment group.

There were several limitations in this study. Firstly, the sample size was small, with only eight studies included, and the outcome unit was not uniform. Therefore, only the standardized mean difference could be used for a comparison. Secondly, the intervention programs that were adopted were not uniform, with both a single WBVT program and a combination of WBVT and other intervention programs being used. Finally, this study was a single-arm meta-analysis without a control group due to insufficient RCT studies. At present, the mechanism by which WBVT increases lean mass and bone minerals remains uncertain, and further studies are required. In addition, if sufficient RCT studies can be included, a control group can be added for a further meta-analysis. Furthermore, WBVT may have various effects on people of different age groups with different diseases, with future studies potentially focusing on comparing the impacts of WBVT on different populations.

## 5. Conclusions

WBVT can improve femur bone density, total body bone mineral content, and lean mass in children and adolescents with motor disabilities. However, it does not have any impact on the bone density of the lumbar spine. It can be used as a potential program to improve bone minerals in children and adolescents with motor disabilities.

## Figures and Tables

**Figure 1 children-09-00266-f001:**
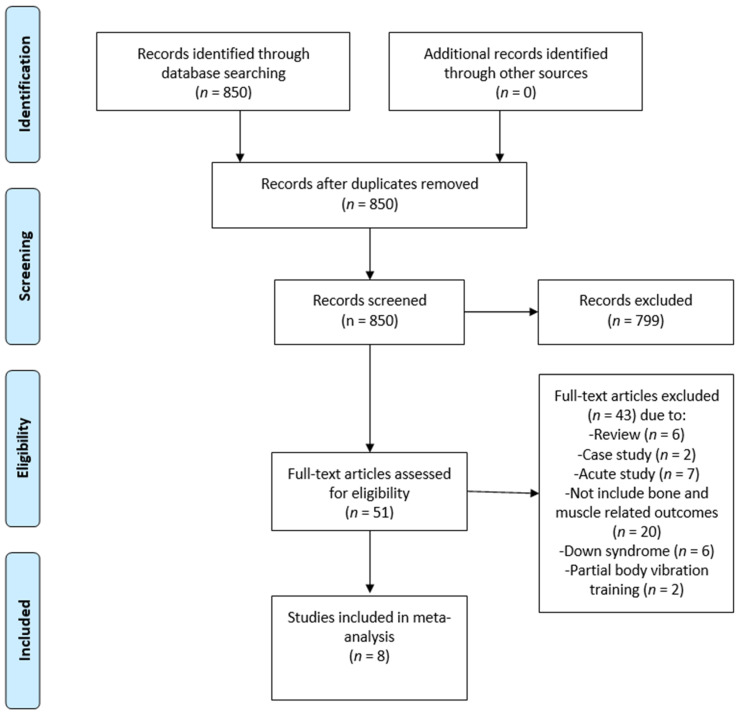
Flow diagram of the search results using the Preferred Reporting Items for Systematic Reviews and Meta-Analysis (PRISMA).

**Figure 2 children-09-00266-f002:**
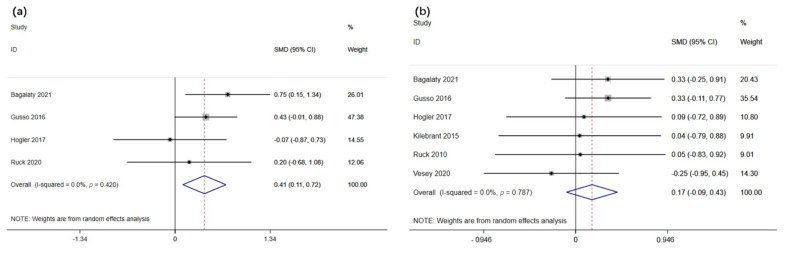
Forest plot illustrating the effects of whole-body vibration training on (**a**) femur and (**b**) lumbar spine bone mineral density in children and adolescents with motor disabilities.

**Figure 3 children-09-00266-f003:**
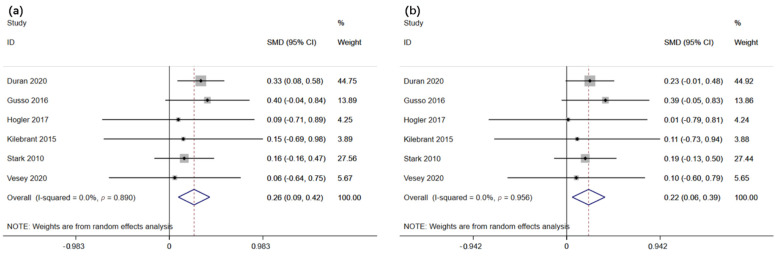
Forest plot illustrating the effects of whole-body vibration training on (**a**) total body bone mineral content and (**b**) lean mass in children and adolescents with motor disabilities.

**Figure 4 children-09-00266-f004:**
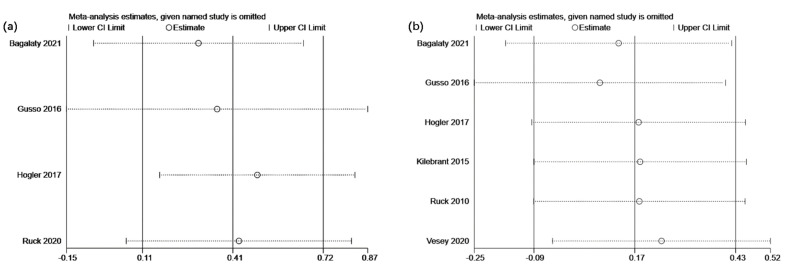
Sensitivity analyses illustrating the effects of whole-body vibration training on (**a**) femur and (**b**) lumbar spine bone mineral density in children and adolescents with motor disabilities.

**Figure 5 children-09-00266-f005:**
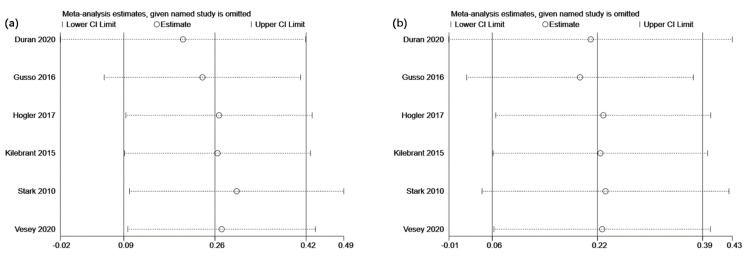
Sensitivity analysis illustrating the effects of whole-body vibration training on (**a**) total body bone mineral content and (**b**) lean mass in children and adolescents with motor disabilities.

**Figure 6 children-09-00266-f006:**
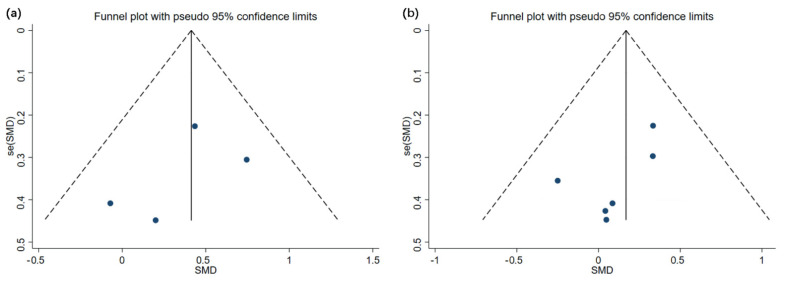
Funnel plots of publication bias for (**a**) femur and (**b**) lumbar spine bone mineral density in whole-body vibration training.

**Figure 7 children-09-00266-f007:**
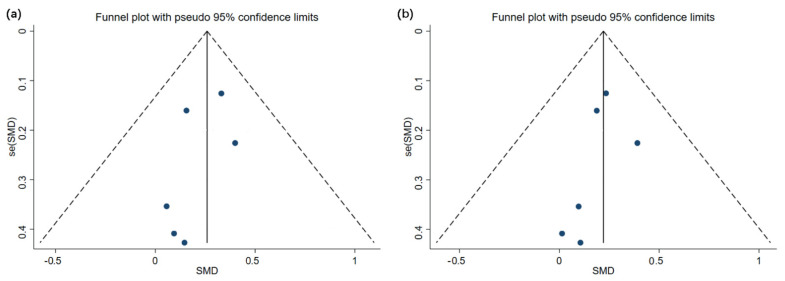
Funnel plots of publication bias for (**a**) total body bone mineral content and (**b**) lean mass in whole-body vibration training.

**Table 1 children-09-00266-t001:** Characteristics of the included studies.

Study	Type	Age (y)	Gender	Disease	Duration	WBVT Program	Index	DXA Device
Bagalaty 2021	RCT	5.0 ± 0.84	10 M/13 F	CP	12 weeks;3 x/week	5–25 Hz; OA = 0–3.9 mm; Erect position, squatting, and side-standing for 20 min; Side-alternating vibrations.	Lumbar-spine BMD; Femur BMD	DP3, Lunar Corporation, USA
Duran 2020	Cohort	11.9 ± 2.7	76 M/52 F	CP	6 months; 3–10 x/week	8–20 Hz; OA = 1–2 mm; Standing on a 40 degrees inclined pedal and train for 15 minutes; Side-alternating vibrations.	Total-body BMC; Lean mass	GE, UK
Gusso 2016	Cohort	16.2	23 M/17 F	CP	20 weeks; 4 x/week	15–20 Hz; OA = 1 mm; Stand on a vibration pedal with knees bent and train for 9 min; Side-alternating vibrations.	Lumbar-spine BMD; Femur BMD; Total-body BMC; Lean mass	GE Lunar Prodigy, USA
Hogler 2017	RCT	10.5 ± 2.9	6 M/6 F	OI	5 months; 2 x/day	20–25 Hz; OA = 2–6 mm; Stand on a 10–45 degrees inclined vibration pedal with knees bent and train for 18 min; Side-alternating vibrations.	Lumbar-spine BMD; Femur BMD; Total-body BMC; Lean mass	GE Lunar Prodigy, USA
Kilebrant 2015	Cohort	5.1–16.3	4 M/7 F	MD	6 months; 2 x/week	40–42 Hz; OA = 0.2 mm; Stand on a vibration pedal with knees bent and train for 5–15 min; Vertical vibrations.	Lumbar-spine BMD; Total-body BMC; Lean mass	GE Lunar Prodigy, USA
Ruck 2010	RCT	8.2 ± 0.9	8 M/2 F	CP	6 months; 5 x/week	12–18 Hz; OA = 4 mm; Stand on a 35 degrees inclined vibration pedal and train for 15 min; Side-alternating vibrations.	Lumbar-spine BMD; Femur BMD	QDR Discovery, Hologic Inc, USA
Stark 2010	Cohort	9.76	44 M/34 F	CP	6 months; 1 x/day	5–25 Hz; OA = 0–3.9 mm; Stand on a vibration pedal and train for 15 min; Side-alternating vibrations.	Total-body BMC; Lean mass	GE, Germany
Vesey 2020	Cohort	15.7 ± 2.9	8 M/8 F	MD	20 weeks; 4 x/week	12–20 Hz; OA = 1 mm; Stand on a vibration pedal and train for 15 min; Side-alternating vibrations.	Lumbar-spine BMD; Total-body BMC; Lean mass	GE Lunar Prodigy, USA

RCT = randomized controlled trial; M = male; F = female; BMD = bone mineral density; BMC = bone mineral content; WBVT = whole body vibration training; CP = cerebral palsy; OI = osteogenesis imperfecta; MD = musculoskeletal disorders; OA = oscillation amplitude.

**Table 2 children-09-00266-t002:** Depiction of the RCT quality and bias assessment.

PEDro Scale		1	2	3	4	5	6	7	8	9	10	11	Total	LE	LR
Bagalaty 2021		Y *	Y	N	Y	N	N	N	Y	Y	Y	Y	6/10	1B	B
Hogler 2017		Y *	Y	Y	N	N	N	N	Y	Y	Y	Y	6/10	1B	B
Ruck 2010		Y *	Y	Y	Y	N	N	N	N	Y	Y	Y	6/10	1B	B
**NIH scale**	**1**	**2**	**3**	**4**	**5**	**6**	**7**	**8**	**9**	**10**	**11**	**12**	**Total**		
Duran 2020	Y	Y	Y	N	Y	N	N	N	Y	Y	Y	NA *	7/11	2B	C
Gusso 2016	Y	Y	Y	Y	Y	Y	N	N	Y	Y	Y	NA *	9/11	2B	C
Kilebrant 2015	Y	Y	Y	Y	Y	Y	N	N	N	Y	Y	NA *	8/11	2B	C
Stark 2010	Y	Y	Y	Y	Y	Y	N	N	Y	Y	Y	NA *	9/11	2B	B
Vesey 2020	Y	Y	Y	Y	Y	Y	N	N	Y	Y	Y	NA *	9/11	2B	B

Y = yes; N = no; NA = not applicable; NR = not reported. * = not included in total score; LE = level of evidence; LR= level of recommendation.

## Data Availability

The data that support the findings of the study are available from the corresponding author upon reasonable request.

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
