# Peer review of "The Impact of Whole-Body Vibration Training on Bone Minerals and Lean Mass in Children and Adolescents with Motor Disabilities: A Systematic Review and Meta-Analysis"

_children, 2022, doi:10.3390/children9020266_

Round 1
Reviewer 1 Report
The systematic review is very well chained, respecting all the steps recommended by the PRISMA. I congratulate the authors for the excellent work and bring minimum scores that I considered relevant to be seen by the authors.
Line 70-72- I suggest to declare that this systematic review and MA followed the recommendations proposed by The Preferred Reporting Items for Systematic Reviews and Meta-Analyses (PRISMA).
Line 73-82 - Has the PECOS strategy been established? If so, it's worth reporting.
Line 77- The search strategy and results for every database should be present in the text, or perhaps as a table, allowing readers to follow the entire review process.
Author Response
Responses to reviewers
Reviewer 1
|
Comment |
Response |
1. |
The systematic review is very well chained, respecting all the steps recommended by the PRISMA. I congratulate the authors for the excellent work and bring minimum scores that I considered relevant to be seen by the authors. |
Thank you. |
2. |
Line 70-72- I suggest to declare that this systematic review and MA followed the recommendations proposed by The Preferred Reporting Items for Systematic Reviews and Meta-Analyses (PRISMA). |
It has been supplemented in the method section.(Lines 87-89) |
3. |
Line 73-82 - Has the PECOS strategy been established? If so, it's worth reporting. |
Thank you for your suggestion. We have reported PECOS.(Lines 94-97) |
4. |
Line 77- The search strategy and results for every database should be present in the text, or perhaps as a table, allowing readers to follow the entire review process. |
I agree with you. I have added the search strategy to the appendix. |
Reviewer 2 Report
In my opinion, this is a manuscript of great interest to the scientific community, but there are some considerations that I believe should be taken into account for the manuscript to be accepted:
1. In my opinion it is necessary to include in the tables of the manuscript all the necessary information in this type of study, in this sense I strongly recommend following some guide about this. In particular, I believe that the following would be very useful https://www.ncbi.nlm.nih.gov/pmc/articles/PMC8533415/
2. In the manuscript there is hardly any information on the type of platform on which the whole body vibration training was performed, this is essential since the training parameters vary greatly depending on the type of vibration (tilting, vertical), if all the parameters recommended in the previous guide are included in the tables this will be remedied.
3. The study includes information on the risk of bias through the PEDro scale, but does not include the quality of evidence, I suggest incorporating the GRADE system (you can consult what it is in this article, for example, https://www.elsevier.es/en-revista-cirugia-espanola-english-edition--436-articulo-grade-system-classification-quality-evidence-S2173507714000908).
4. I think that the level of evidence, the level of conclusion, and the level of recommendation should be included in the manuscript, I think this would be very useful. (e.g., https://www.ncbi.nlm.nih.gov/pmc/articles/PMC3124652/)
5. Why has a funnel plot been included with so few studies, do you think this is appropriate?
6. I think the manuscript should conclude with the level of evidence, conclusion, and recommendation of WBV on bone mineral and lean mass in the study population.
Author Response
Responses to reviewers
Reviewer 2
|
Comment |
Response |
1. |
In my opinion it is necessary to include in the tables of the manuscript all the necessary information in this type of study, in this sense I strongly recommend following some guide about this. In particular, I believe that the following would be very useful https://www.ncbi.nlm.nih.gov/pmc/articles/PMC8533415/ |
Thank you very much for your suggestion. This knowledge will help us improve the quality of manuscripts. It also provides guidance for our future vibration training research. |
2. |
In the manuscript there is hardly any information on the type of platform on which the whole body vibration training was performed, this is essential since the training parameters vary greatly depending on the type of vibration (tilting, vertical), if all the parameters recommended in the previous guide are included in the tables this will be remedied. |
The information in the relevant table has been supplemented. (Table 1) |
3. |
The study includes information on the risk of bias through the PEDro scale, but does not include the quality of evidence, I suggest incorporating the GRADE system (you can consult what it is in this article, for example, https://www.elsevier.es/en-revista-cirugia-espanola-english-edition--436-articulo-grade-system-classification-quality-evidence-S2173507714000908). |
Thank you for your suggestion. I tried to use gradepro to evaluate the quality of evidence, but failed to complete it. This study is a single arm study, which only included the experimental group, not the control group, or the control group is inconsistent. Therefore, it is difficult to use gradepro system for quality evaluation. The quality evaluation scheme and research methods used in this study refer to previous articles on similar situations. Please check.
Suárez-Iglesias D, Miller KJ, Seijo-Martínez M, Ayán C. Benefits of Pilates in Parkinson’s Disease: A Systematic Review and Meta-Analysis. Medicina (B Aires). 2019;55(8):476. doi:10.3390/medicina55080476
Li, S., Ng, W.H., Abujaber, S. et al. Effects of resistance training on gait velocity and knee adduction moment in knee osteoarthritis patients: a systematic review and meta-analysis. Sci Rep 11, 16104 (2021). https://doi.org/10.1038/s41598-021-95426-4 |
4. |
I think that the level of evidence, the level of conclusion, and the level of recommendation should be included in the manuscript, I think this would be very useful. (e.g., https://www.ncbi.nlm.nih.gov/pmc/articles/PMC3124652/) |
I agree with you. I have added the evidence level and recommendation level in Table 2. (And Lines 109-117; Lines 149-151) |
5. |
Why has a funnel plot been included with so few studies, do you think this is appropriate? |
Theoretically, funnel plots are most suitable for more than 10 studies. However, previous studies have used funnel plots in the case of low sample size. Although the accuracy of this evaluation is not enough, it may also reflect bias to some extent. In this way, readers can have a more comprehensive understanding of the information of this research.
Lu Y, Wang W, Ding X, Shi X. Association between the promoter region of serotonin transporter polymorphisms and recurrent aphthous stomatitis: A meta-analysis. Arch Oral Biol. 2020;109:104555. doi:10.1016/j.archoralbio.2019.104555 |
6. |
I think the manuscript should conclude with the level of evidence, conclusion, and recommendation of WBV on bone mineral and lean mass in the study population. |
Thank you for your suggestion. We have supplemented it in the discussion section. (Lines 221-222) |
Round 2
Reviewer 2 Report
The authors have made the recommended changes. I have no further comments.